# Liquid Biopsy for Disease Monitoring in Non-Small Cell Lung Cancer: The Link between Biology and the Clinic

**DOI:** 10.3390/cells10081912

**Published:** 2021-07-28

**Authors:** Maria Gabriela O. Fernandes, Catarina Sousa, Joana Pereira Reis, Natália Cruz-Martins, Conceição Souto Moura, Susana Guimarães, Ana Justino, Maria João Pina, Adriana Magalhães, Henrique Queiroga, José Agostinho Marques, José Carlos Machado, José Luís Costa, Venceslau Hespanhol

**Affiliations:** 1Pulmonology Department, Centro Hospitalar Universitário de São João, Alameda Prof. Hernâni Monteiro, 4200-319 Porto, Portugal; catarinacdsousa@gmail.com (C.S.); adrimagalhaes08@gmail.com (A.M.); hjqueiroga@gmail.com (H.Q.); marquesa@med.up.pt (J.A.M.); hespanholv@gmail.com (V.H.); 2Faculty of Medicine, University of Porto, Alameda Prof. Hernani Monteiro, 4200-319 Porto, Portugal; ncmartins@med.up.pt (N.C.-M.); susana.melo.gui@gmail.com (S.G.); josem@ipatimup.pt (J.C.M.); jcosta@ipatimup.pt (J.L.C.); 3Institute of Molecular Pathology and Immunology of the University of Porto (IPATIMUP), 4200-135 Porto, Portugal; joanar@ipatimup.pt (J.P.R.); anarjustino@gmail.com (A.J.); mpina@ipatimup.pt (M.J.P.); 4Institute for Research and Innovation in Health (i3S), University of Porto, Rua Alfredo Allen, 4200-135 Porto, Portugal; 5Pathology Department, Centro Hospitalar Universitário de São João, Alameda Prof. Hernâni Monteiro, 4200-319 Porto, Portugal; moura.conceicao@gmail.com

**Keywords:** lung cancer, adenocarcinoma, liquid biopsy, cell-free DNA, tumour-free DNA, next-generation sequencing, clinical outcomes

## Abstract

Introduction: Cell-free DNA (cfDNA) analysis offers a non-invasive method to identify sensitising and resistance mutations in advanced Non-Small Cell Lung Cancer (NSCLC) patients. Next-generation sequencing (NGS) of circulating free DNA (cfDNA) is a valuable tool for mutations detection and disease′s clonal monitoring. Material and methods: An amplicon-based targeted gene NGS panel was used to analyse 101 plasma samples of advanced non-small cell lung cancer (NSCLC) patients with known oncogenic mutations, mostly EGFR mutations, serially collected at different clinically relevant time points of the disease. Results: The variant allelic frequency (VAF) monitoring in consecutive plasma samples demonstrated different molecular response and progression patterns. The decrease in or the clearance of the mutant alleles was associated with response and the increase in or the emergence of novel alterations with progression. At the best response, the median VAF was 0% (0.0% to 3.62%), lower than that at baseline, with a median of 0.53% (0.0% to 9.9%) (*p* = 0.004). At progression, the VAF was significantly higher (median 4.67; range: 0.0–36.9%) than that observed at the best response (*p* = 0.001) and baseline (*p* = 0.006). These variations anticipated radiographic changes in most cases, with a median time of 0.86 months. Overall, the VAF evolution of different oncogenic mutations predicts clinical outcomes. Conclusion: The targeted NGS of circulating tumour DNA (ctDNA) has clinical utility to monitor treatment response in patients with advanced lung adenocarcinoma.

## 1. Introduction

The advances stated in identifying actionable driver mutations and resistance mechanisms to target therapies have markedly changed the clinical approach in lung cancer treatment and have primarily contributed to improve the survival of patients with non-small-cell lung cancer (NSCLC) [1]. Additionally, the increasing number of possible targets and treatment options have made tumour genotyping a significant step in the decision process. Target therapies for patients harbouring mutations in the Epidermal Growth Factor Receptor (EGFR), Anaplastic Lymphoma Kinase (ALK), ROS Proto-Oncogene 1 (ROS1), B-Raf Proto-Oncogene (BRAF), Erb-B2 Receptor Tyrosine Kinase 2 (ERBB2), MET proto-oncogene, receptor tyrosine kinase (MET), Neurotrophic Receptor Tyrosine Kinase (NTRK), and Kirsten Rat Sarcoma Viral Oncogene Homolog (KRAS) genes are available, while other genetic alterations are the focus of promising clinical trials [2]. Moreover, associated with target therapies, multiple molecular resistance mechanisms have been pointed out, and the detection of these alterations has implications in selecting further treatments, as demonstrated for EGFR and ALK patients.

Tissue biopsy is the preferred source of tumour DNA, being the current standard for lung cancer genotyping and detection of resistance mechanisms [2]. However, tissue is usually obtained by invasive methods and provides small amounts of DNA, besides giving a static snapshot of the entire disease that does not fully reflect the tumour’s temporal and spatial heterogeneity. In this way, liquid biopsies have been pointed at with increasing interest given their non-invasive, low-risk, and less expensive nature compared to tissue biopsies; their capability of detecting genetic alterations in cancer patients; and potentiality for longitudinal molecular disease monitoring. It is noteworthy that using liquid biopsies to detect resistance mechanisms to targeted therapies and to monitor the disease′s evolution is an area of growing interest, especially the analysis of circulating tumour DNA (ctDNA) [3].

Circulating tumour DNA originates from the shedding of tumour-derived double-strand DNA fragments into the blood, released by tumour cells that undergo necrosis and apoptosis [3]. Digital polymerase chain reaction (PCR)-based tests are the most used to detect ctDNA mutant variants, revealing good sensitivity and specificity, particularly in EGFR mutant lung cancer [4,5]. Currently, the next-generation assays are being applied to liquid biopsies since they can simultaneously detect multiple alterations, potentially capturing the tumour heterogeneity. Thus, a liquid biopsy provides a rapid approach to address the therapeutic response to target therapies, with important implications in managing lung cancer patients for whom tissue biopsies are difficult to obtain. In addition, plasma molecular indicators of progression may anticipate radiographic and clinical progression. In this sense, in this prospective case-series report, we aim to demonstrate the usefulness of monitoring ctDNA in NSCLC patients with various oncogenic alterations, mostly EGFR mutations, and detectable ctDNA and the corresponding relationship with clinical outcomes.

## 2. Materials and Methods

From a prospective cohort of untreated patients with advanced lung adenocarcinoma, selected for the detection of oncogenic mutations in cfDNA and longitudinal molecular monitoring, illustrative cases with oncogenic mutations detected in cfDNA were included in this analysis and were stratified according to the type of mutation detected. A study diagram is available in Appendix A.

Plasma samples were serially collected and sequenced around three distinct time points, i.e., baseline, best response, and progression. Targeted plasma NGS was performed using a validated amplicon-based NGS Oncomine™ Lung cfDNA Assay (Thermo Fisher Scientific, Waltham, MA, USA) that uses target gene enrichment using PCR with a set of primers for exons or hotspots of the selected gene, covering more than 150 hotspots on ALK, BRAF, EGFR, ERBB2, KRAS, Mitogen-Activated Protein Kinase Kinase 1 (MAP2K1), MET, NRAS Proto-Oncogene, GTPase (NRAS), Phosphatidylinosi-tol-4,5-Bisphosphate 3-Kinase Catalytic Subunit Alpha (PIK3CA), ROS1, and Tumour Protein P53 (TP53). The baseline ctDNA was compared to pre-treatment tumour samples sequenced with the Ion AmpliSeq Colon and Lung Cancer Research Panel v2 (Ion Torrent, Waltham, MA, USA) used to detect DNA changes in a multiplex PCR-based test that analyses 1850 hotspots and targeted regions in 22 genes.

Blood samples were collected in K2EDTA BD Vacutainer^®^ PPT™ Plasma Preparation Tubes (Becton Dickinson, Franklin Lakes, NJ, USA). DNA was extracted with the MagMax Cell-Free Total Nucleic Acid Isolation Kit (Thermo Fisher Scientific) and quantified with the dsDNA HS assay kit using a Qubit 3.0 or 4.0 Fluorometer (Thermo Fisher Scientific). The test sensitivity for detection of both single nucleotide variants (SNVs) and short indels is down to 0.1% LOD. Sequencing and bioinformatic analysis are detailed in Appendix A.

Tumour staging was based on the 8th edition from January 2018. The tumour, node, metastasis (TNM) staging of patients included until December 2017 was reclassified with the 8th edition [6,7]. Tumour size (T) was measured as the longest diameter of the primary lesion and adenopathies (N) as the short axis assessed using a CT scan [8].

All subjects gave their informed consent for the study′s inclusion. The study was conducted in accordance with the Helsinki Declaration, and the study protocol (CES-108/14) was approved by the Ethics Committee of the Centro Hospitalar e Universitário de São João (CHUSJ), Porto-Portugal.

Most statistical analyses were descriptive, with categorical data presented as absolute (n) and relative frequencies, continuous variables as medians, interquartile ranges (IQR), and minimum and maximum values, when appropriate. The non-parametric Wilcoxon and Kruskal–Wallis tests were used to determine the differences in VAF concentration at the different time points considered. All statistical analyses were conducted using the Statistical Package for Social Sciences (SPSS, IBM Corp, Chicago, IL, USA) software, version 25.0, with an alpha set at 0.05.

## 3. Results

A total of 101 blood samples of 13 patients with stage IV adenocarcinoma were analysed (median of 6, from 3 to 21 samples per patient). Ten patients had EGFR sensitising mutations, two BRAF p.(V600E), and one patient with a combination of KRAS/TP53/STK11 mutations. Seven (53.8%) patients were females, and six were males, with a median age of 60 years old (range: 38 to 76), most (76.9%) were non-smokers. The patients′ clinical and molecular features are detailed in Appendix A.

### Sequencing Results and Correlation between Molecular Monitoring and Clinical Outcomes

One hundred and one plasma samples were sequenced, corresponding to 12 patients with oncogenic alterations detected in cfDNA (“ctDNA positive patients”) and 1 patient without detectable alterations (“ctDNA negative patient or non-shedder”). Plasma molecular ctDNA results were similar to those of the paired tissue DNA.

At baseline, the cfDNA concentration ranged from 0.28 to 17.70 µg/µL, with a median of 1.55 µg/µL. A median VAF of 0.53% (range: 0% to 9.90%) was detected for the original mutation in plasma, being significantly inferior to that of tissue (median 13.0; range: 0–96.5%; *p* < 0.001) (Figure 1A). The correlation between the cfDNA concentration and the VAF was not statistically significant (*r* = −0.319; *p* = 0.289) (Figure 1B).

The following three clinical time points were considered for monitoring purposes: baseline, at the best response, and progression. To capture the treatment′s selective pressure, the identified mutations detected at baseline in cfDNA were subsequently analysed to identify changes in the allelic frequency (VAF). Globally, we found that at the best response, the VAF was lower than that at the baseline, with a median of 0% (0% to 3.62%) and 0.53% (0% to 9.9%), respectively (*p* = 0.004). On the other hand, at progression, the VAF was significantly higher (median 4.67; from 0% to 36.9%) than that observed at the best response time point (*p* = 0.001) and even that at the baseline (*p* = 0.006) (Figure 2A). Looking at cfDNA concentration, there was a decrease at the best response and an increase at progression (Figure 2B).

Thus, the following three patterns were defined: VAF decrease, VAF increase, and no VAF change. Additionally, the emergence of new alterations was described. The decrease and/or disappearance in the cfDNA of a mutation was associated with radiographic and clinical response, expressed as a partial response or stable disease, as shown in most cases, except for the ctDNA negative patient #89. On the other hand, the increases in the VAF of these mutations were associated with progression in most patients (11/13, 84.6%,) and two patients had no detectable alterations, the “negative” #89 and #65 without detectable changes of ctDNA at progression (Table 1).

At baseline, among the EGFR mutant patients, the VAF ranged from 0.0% to 5.50%, with a median of 0.65%. Sensitising mutations were exon 19 deletions and L785R in eight and two cases, respectively. The T790M mutation was the resistance mechanism to the first- or second-generation tyrosine kinase inhibitors (TKIs) in 60% (6/10) of cases (Table 1 and Figure 3).

When present at the progression time point, the T790M VAF varied from 0.65% to 12.5%, with a median of 3.06%. The allelic fraction of the T790M mutation was lower than that of the sensitising one (median: 6.8%; range: 0.13–27.6%; *p* = 0.075) (Figure 3F), suggesting its sub-clonality, with a median EGFR sensitising/T790M ratio of 2.04 (range: 0.89–3.9). In only one case, the T790M VAF exceeded the EGFR sensitising VAF (Figure 3B). In this case, the T790M was already present at diagnosis. Regarding progression on EGFR TKIs, we observed that both the rising levels of the original sensitising mutation with (Figure 3) or without (Figure 4) the emergence of the resistance T790M mutation were linked to clinical progression.

The levels of sensitising and resistance mutations evolved parallelly and anticipated clinical and radiological progression with a median time of 0.86 months (range: −3.1 before to +6.2 months after clinical progression). Here, one detected resistance mechanism to osimertinib was the increase in the driver mutation and the emergence of a de novo C797S mutation in the EGFR tyrosine kinase domain (Figure 5). In two of the EGFR patients included, along with the disease′s course, there were no detectable mutations regardless of the presence and quantity of cfDNA, representing the absence of mutant variants, and eventually non-shedding DNA from the tumour (Figure 6). Six patients with EGFR performed a tissue re-biopsy at progression. Plasma and tissue were concordant in 5/6 (one negative ctDNA/negative tissue DNA and four positive ctDNA/positive tissue DNA) and discordant in one (negative ctDNA/positive tissue DNA, case#65).

One patient with a combination of KRAS/TP53/STK11 mutations was included. The three clones had a synchronous and proportional variation, supporting the same tumoral origin (Figure 7). In both BRAF-V600E cases, mutation response and progression were demonstrated by a reduction and an increase in the VAF, respectively (Figure 8). It is worth emphasising that patient #80 developed leptomeningeal progression, and this event did not translate into a corresponding VAF increase (Figure 8B).

## 4. Discussion

In this prospective cohort, we explored the usefulness of ctDNA longitudinal monitoring in different NSCLC settings, evaluating variations in the allelic frequency of mutated genes or observing the emergence of secondary mutations. Both quantitative and qualitative molecular plasma findings were correlated with clinical and radiographic data at the different time points considered. In addition, when available, the plasma findings were compared with the tissue findings, while the sensitivity and concordance of this NGS plasma assay with tissue NGS was already addressed in a previous study [9].

Considering that cfDNA NGS has revealed to be particularly useful to detect many genetic alterations, including alterations not detected in the primary biopsy, its identification will allow for them to be tracked over time with consequent application to various oncogenic scenarios. Here, we demonstrate the importance of cfDNA molecular monitoring to complement clinical surveillance, elucidating the clinical impact of treatment on tumour clonal heterogeneity and, ultimately, the emergence of multiple resistance mechanisms associated with TKIs.

To date, few studies have described the clinical utility of cfDNA for assessing treatment response and resistance patterns. The most illustrative example is the detection of targetable mutations in EGFR-mutated lung cancer patients. Previous studies have proven the feasibility of detecting EGFR mutations in ctDNA [4,5], while predictive markers of response to EGFR tyrosine kinase therapy [10,11] can be used for monitoring EGFR alterations in plasma during the disease [6,12,13]. Plasma ctDNA monitoring allowed for the emergence of T790M to be detected in 40–47% of cases, a median of 2.2 months prior to disease progression [12,13]. In cases of exclusive central nervous system involvement, an alternative method is to look for variations in Cerebrospinal Fluid Cell-Free DNA as a liquid biopsy [7]. With the third generation TKIs, resistance mechanisms are more heterogeneous, with the emergence of several types of alterations being stated over time by plasma assessment [14,15]. For patients treated with second line osimertinib, the paired ctDNA samples analysed with NGS revealed that the loss of the T790M mutation was the most frequent phenomenon. However, other acquired EGFR mutations have also been found, with C797S mutations being the most common. MET amplification, HER2 and PIK3CA amplification, RET and NTRK1 fusions, BRAF V600E mutations, and acquired cell cycle gene alterations can also be present [16]. Additionally, changes in ctDNA may have a prognostic value, where plasma clearance has been linked to better outcomes [17,18,19]. Conversely, the ctDNAs′ persistence can be associated with a lower ORR. The molecular non-responders exhibited limited changes in post-treatment ctDNA levels and experienced significantly shorter PFS, detected, on average, 4 weeks before CT imaging [20]. “Molecular” progression is generally detected prior to radiographic progression, with this being noticed with the emergence of T790M in different studies [13,21,22].

In our case series, in the EGFR scenario, we observed that when progression occurs, a rise in the sensitive EGFR mutations accompanies the emergence of resistance alterations, such as T790M or C767S in cis in case #74. One of the advantages of analysing ctDNA with NGS is to observe the correlation between the EGFR activating mutation and T790M and its association with clinical progression. Additionally, repeated tissue biopsies have validated the data found in plasma, confirming the results obtained and proving that liquid biopsy is a surrogate of tissue biopsy. These data highlight the advantage of using NGS-based plasma analysis, indispensable to face the complexity of molecular changes that emerge after the third generation TKIs.

Regarding the BRAF V600E mutation in NSCLC patients, few reports are available in the literature [23], and we observed that, in this setting, ctDNA is helpful for monitoring disease outcomes, as in the EGFR setting. We have consistently observed that during the disease, after treatment, there is a decrease in the amount of mutant DNA assessed by the allele frequency with sustained DNA clearance mirroring the clinical response, followed by an increase in ctDNA when progression occurs. In some cases, multiple mutations are identified that generally follow a similar trend in serial samples, confirming their clonality. In EGFR patients, TP53 and KRAS mutations coexist, albeit in minimal amounts and not impacted, as expected, by targeted therapies. Furthermore, and even when no targeted therapies are available, the surveillance of the mutant variants detected in plasma helps to assess the response to non-specific therapies, including chemotherapy (case #15) and immunotherapy, as occurs in KRAS patients treated with immune checkpoint inhibitors.

On the other hand, a good correlation between plasma DNA changes, tissue findings, and clinical evolution was found in this series. Indeed, with ctDNA analysis, it was possible to monitor the response to target therapy, detect resistance variants, and, in some cases, anticipate progression, which would not be possible with repeated tissue biopsies. Moreover, the changes in therapy correlated adequately with the changes observed in ctDNA. Additionally, cfDNA monitoring allows the molecular visualisation of tumour clonal dynamics, where the difference between the clones can explain the mixed radiographic responses sometimes observed. New perspectives of interpreting ctDNA dynamic chances are being explored. The combination of qualitative and quantitative data expressed by the VAF, in combination with time as a function, gathered with mathematical modelling, is currently being tested with exciting results [24].

However, liquid biopsies have some drawbacks, particularly related to the lack of sensitivity, the absence of the release of tumour DNA into circulation, low-burden disease, and the inability to look into histologic changes. As we observed in cases #89 and #65, sometimes monitoring did not contribute to understanding the disease′s evolution. A careful interpretation of null plasmatic findings must be performed, and clinical and radiographic should not be neglected, as the central nervous system and oligo-progression may not translate into measurable changes in ctDNA (#80 and #89).

Considering these limitations, we cannot ignore the importance of tissue biopsies and the potential of different biological fluids in the analysis of ctDNA. In fact, it is unclear and has not been prospectively validated if ctDNA findings may be enough to support therapeutic changes. Treatment beyond RECIST progression has been shown to improve patients′ outcomes [25]; therefore, treatment discontinuation based solely on plasma findings suggesting “molecular progression” is controversial and not advisable. Further prospective studies to evaluate the impact of molecular monitoring on clinical outcomes are awaited and shall be conducted in the future. Expressly, considering the limitation of acquiring serial tissue sampling and the growing evidence on the reliability of liquid biopsies, this assay should be incorporated into clinical practice as a complementary tool to tissue biopsies and interpreted in conjunction with radiographic and clinical findings. Nevertheless, the analysis of NGS ctDNA gives an anticipated and comprehensive view of the tumour and the heterogeneity of resistance mechanisms present in an individual patient, contributing to the therapeutic personalisation and avoiding risks for the patient.

## 5. Conclusions

This prospective cohort sheds light on the role of cfDNA analysis as a disease monitoring tool in lung cancer patients. It is proof of the concept that the ctDNA dynamics of clonal and sub-clonal, sensitising, and resistance mutations correlate with clinical evolution and outcomes. Targeted NGS-based assays for plasma genotyping allow for the dynamic monitoring with longitudinal follow-up of altered genes and the detection of the emergence of genetic resistance mechanisms during therapy, being an add-tool for treatment decision-making.

## Figures and Tables

**Figure 1 cells-10-01912-f001:**
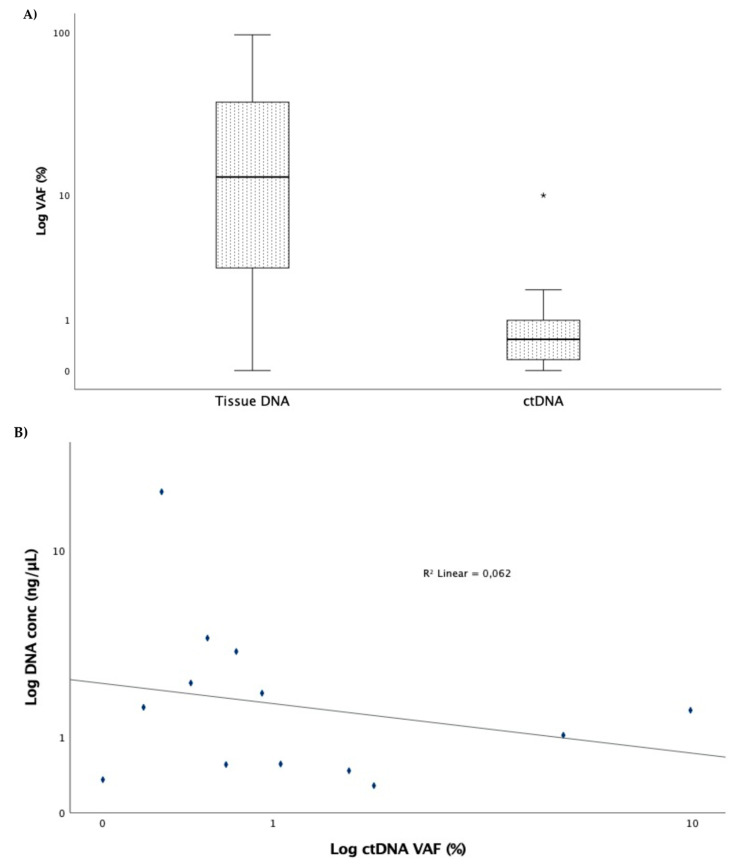
(**A**) Variant allelic fraction between tissue and ctDNA at baseline. Box represents the interquartile range of values and whiskers the median and the 25th and 75th percentile values (* *p* < 0.001, Mann–Whitney test). (**B**) Correlation between ctDNA concentration and variant allelic fraction (*r* = −0.319; *p* = 0.289).

**Figure 2 cells-10-01912-f002:**
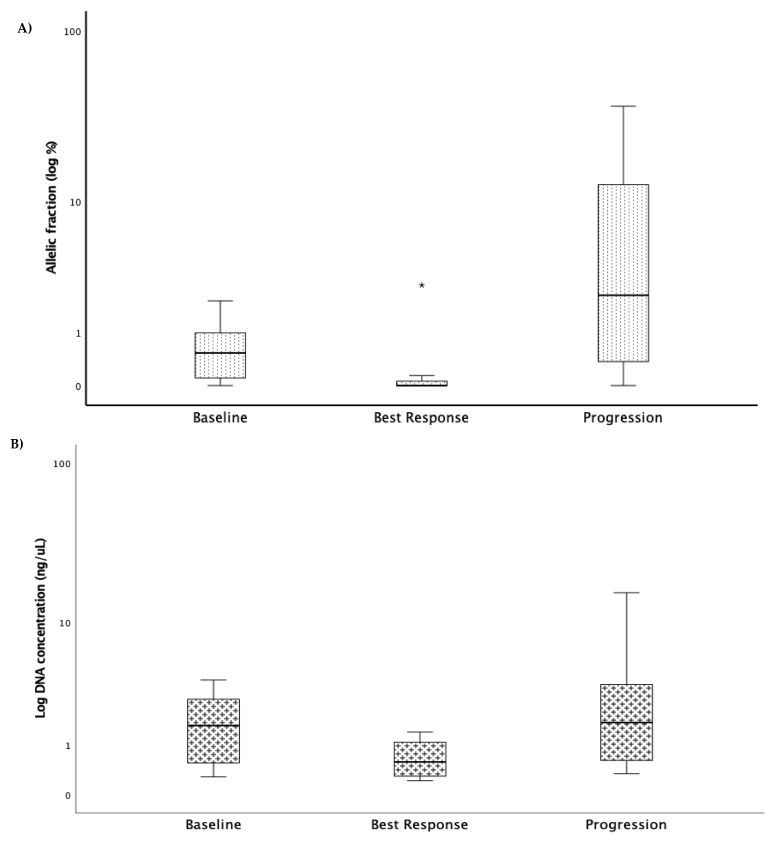
Box plots depicting the detectable variants of the mutated genes (**A**) and cfDNA concentration (**B**) at baseline, best response, and progression. Box represents the interquartile range of values and whiskers the median and the 25th and 75th percentile values (*n* = 13, * *p* = 0.001, Kruskal–Wallis test). The allelic fraction decreased to its lowest level at best response to treatment and increased to its highest level when the disease progressed.

**Figure 3 cells-10-01912-f003:**
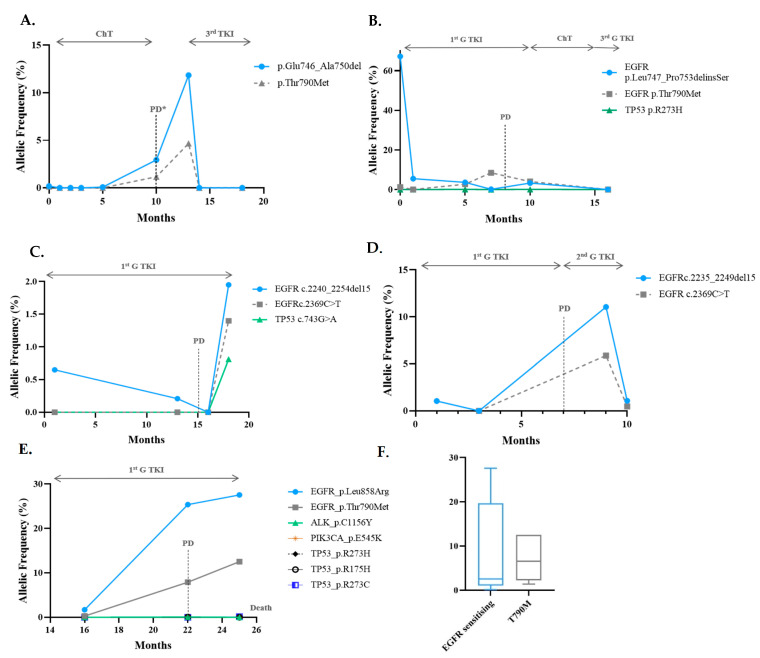
Progression with T790M in EGFR patients. (**A**–**E**), serial monitoring of cfDNA in five patients that developed T790M mutation as resistance mechanism to 1st/2nd G TKIS. The emergence and rise of the T790M (grey line) were accompanied by an increase in the original sensitising mutation (blue line), except for (**B**). In (**B**), dual Del19 and T790M were detected at diagnosis, and when progression occurred, there was a rise of the T790M clone, with a clearance of both clones when 3rd generation TKI was initiated. In (**E**), cfDNA analysis also revealed residual amounts of pathogenic TP53 mutations, ALK C1156Y and PIK3CA E545K, not detected initially. (**F**) Comparison of the VAF ctDNA of the EGFR sensitising mutation and the T790M at progression. The black arrow on the top of the graph indicates the treatment administrated during that time. ChT, chemotherapy; TKI, Tyrosine Kinase Inhibitor; PD, Progressive Disease; PR, Partial Response.

**Figure 4 cells-10-01912-f004:**
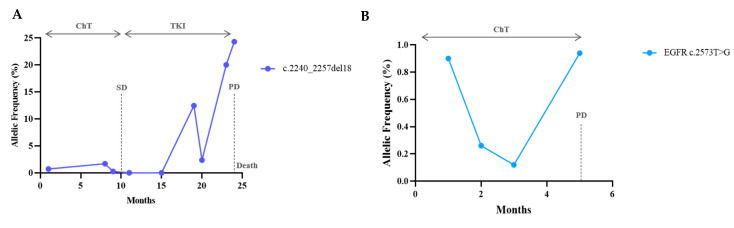
Progression with an increase in the sensitising mutation. (**A**,**B**) Serial ctDNA measurements revealed a VAF decrease with treatment and before radiological progression. There was an increase in the original mutation without other detectable alterations. In (**A**), tissue re-biopsy at 24 months was negative for T790M.

**Figure 5 cells-10-01912-f005:**
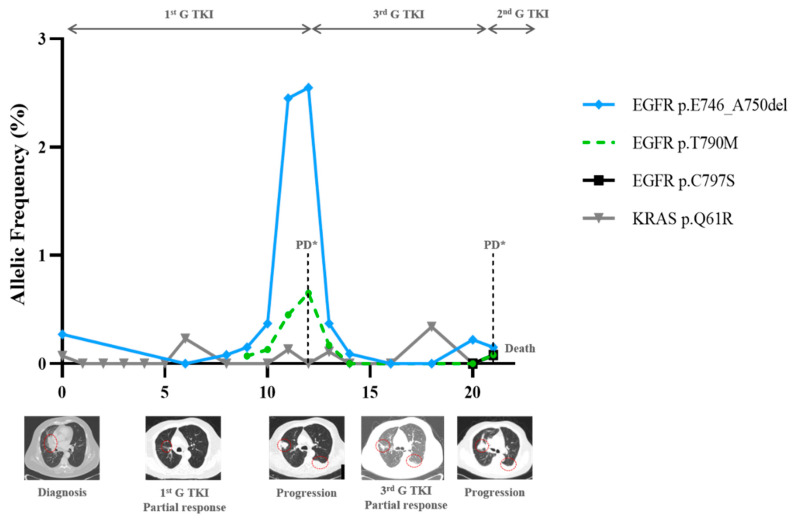
Complete ctDNA monitoring of a patient with Del19 EGFR mutation and representative CT scans at indicated time points. With both 1st and 3rd G TKIs, clinical response was accompanied by molecular decrease in the driver mutation. Clinical progression was preceded by an increase in del19 and the appearance of the T790M. After the initiation of the 3rd G TKI, partial response was observed with ctDNA clearance of both clones. At the recurrence time, the patient lost the T790M, del19 reappeared, and the C767S in cis was detected. Plasmatic alterations at progression were both confirmed with a tissue biopsy represented by *. Red circles refer to main tumour lesions.

**Figure 6 cells-10-01912-f006:**
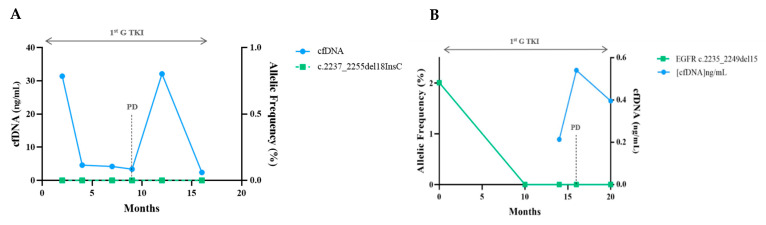
Example of “negative” patients. Despite fluctuations in cfDNA amount in (**A**), no mutations were detected. In (**B**), there was a complete clearance of the driver mutation with response to 1st G TKI, but no change nor a new alteration was stated when progression occurred.

**Figure 7 cells-10-01912-f007:**
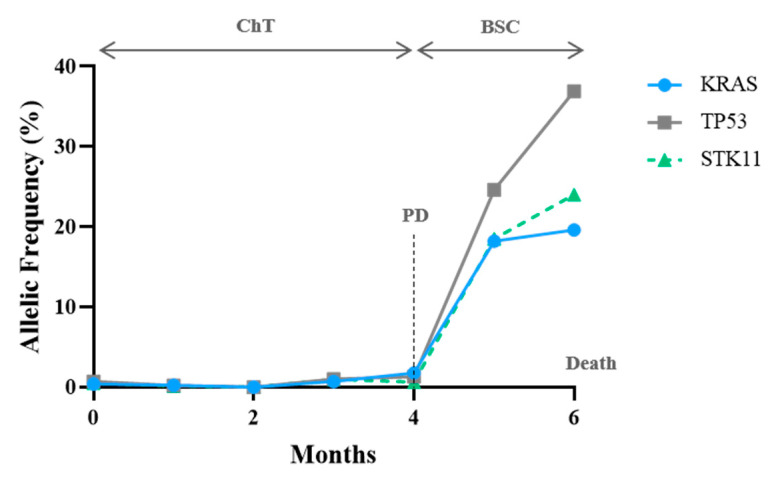
CtDNA monitoring in a KRAS/STK11/P53 mutant patient. At the 3rd cycle of chemotherapy (ChT), there was a proportional decrease in all detected variants. One month before RECIST progressive disease (PD), the patient displayed an increase in all variants, becoming exponential before death occurs.

**Figure 8 cells-10-01912-f008:**
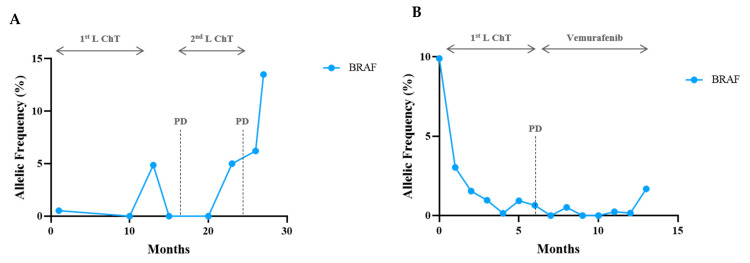
CtDNA monitoring in BRAF 600E mutant patients. In (**A**), 1st line chemotherapy (Cht) with maintenance was stopped due to optic neuropathy. In the treatment-free interval, there was an increase in ctDNA, followed by ctDNA clearance after 2nd line treatment, which was performed until local progression and brain metastasis (24th month). Increase in the BRAF allelic frequency preceded PD in one month. In (**B**), the patient progressed on 1st line chemotherapy with leptomeningeal metastasis at month 4, not accompanied by VAF increase. With vemurafenib, there was a neurological and thoracic improvement with BRAF V600E ctDNA clearance. The patient developed a large pleural effusion and clinical deterioration escorted by ctDNA increase and death.

**Table 1 cells-10-01912-t001:** Patient′s ctDNA NGS sequencing results and progression patterns.

Case	Gender	Age	Genomics	Aa	Tissue DNA VAF%	ctDNA VAF %			Progression Pattern	cfDNA Concentration (µg/µL)			Tissue Rebiopsy at Progression
						Baseline	Best Response	Progression		Baseline	Best Response	Progression	
**2**	F	38	EGFR	c.2240_2257del18	40.00	0.72	0.00	24.30	VAF increase	3.36	1.03	15.60	T790M negative
**15**	M	60	KRAS	c.182A > G	8.80	0.43	0.00	19.60	VAF increase	2.27	31.70	3.17	
			TP53	c.344G > T	25.00	0.71	0.00	36.90					
			STK1	c.597G > T	12.80	0.53	0.00	24.00					
**62**	M	60	BRAF	c.1799T > A	36.30	0.53	0.00	13.50	VAF increase	3.93	0.29	2.22	
**74**	M	59	EGFR	c.2236_2250del15	11.10	0.27	0.00	2.55	VAF increase andde novo T790M	17.70	1.12	0.75	T790M positive
			EGFR	c.2369C > T	0.00	0.00	0.00	0.65					
			KRAS	c.182A > G	0.38	0.07	0.00	0.00					
**80**	F	58	BRAF	c.1799T > A	50.40	9.90	0.14	1.69	VAF increase	1.55	0.86	0.63	
			TP53	c.476C > G	39.20	4.90	0.00	0.00					
**81**	M	40	EGFR	c.2236_2250del15	36.70	0.18	0.00	11.85	VAF increase andde novo T790M	1.62	0.29	1.73	T790M positive
				c.2369C > T	15.00	0.14	0.00	4.67					
**89**	M	64	EGFR	c.2237_2255del18		0.00	0.00	0.00	No change	0.35	0.52	0.34	T790M negative
**91**	F	76	EGFR	c.2573T > G		1.72	0.00	27.55	VAF increase and de novo T790M	0.47		0.95	
				C.2369C > T		0.28	0.00	12.50					
**95**	F	59	EGFR	c.2240_2257del15	96.50	0.65	0.21	1.95	VAF increase and de novo T790M	0.55	0.58	0.59	
				c.2369C > T	0.00	0.00	0.00	1.40					
**107**	F	54	EGFR	c.2240_2257del18	67.20	5.50	3.62	0.13	VAF increase in the T790M	1.03	1.53		
				c.2369C > T	1.25	0.00	2.69	8.46					
**130**	M	63	EGFR	c.2235_2249del15	13.20	1.06	0.00	11.05	VAF increase and de novo T790M	0.56	0.26	4.15	T790M positive
				c.2369C > T	0.00	0.00	0.00	5.88					
**38**	F	65	EGFR	c.2573T > G	6.30	0.91	0.12	1.94	VAF increase	1.98	1.39	24.10	
**65**	F	66	EGFR	c.2235_2249del15	7.60	2.01	0.00	0.00	No change	0.281	0.214	0.396	T790M positive

## Data Availability

The data presented in this study are available on request from the corresponding author.

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
