# Peer review of "Liquid Biopsy for Disease Monitoring in Non-Small Cell Lung Cancer: The Link between Biology and the Clinic"

_cells, 2021, doi:10.3390/cells10081912_

Round 1
Reviewer 1 Report
the manuscript by De Fernadees MGO et al report a prospective study evaluating the therapeutic value of circulating cell DNA mutation detection in lung cancer. This study, although limited to 13 patients with available tissue and blood samples, shows the value of these analyses. For this purpose the authors used an NGS approach allowing the caracterisation of a multitude of targets. Thus, beyond the initial mutations found in the tumour, they can evaluate the disappearance of initial mutations or the appearance of mutations secondary to treatment. Interestingly, the authors indicate that in some cases the characterisation of oncogenic mutations may be a more early indication of progression than imaging. The authors also clearly indicate the limits of this caracterisation and replace these analyses as a new additional element in the patient follow-up system.
Despite the limited number of patients followed, the results are interesting and bring a new light in the management of this disease and therefore deserve to be published with .
Minor révisions
1- In the text it would be useful to specify the sensitivity limit retained by the authors for the analyses with the amplicon-based NGS Oncomine lung cfDNA.
2-The authors detect in fig5 the appearance of the EGFR p.C797S mutation after treatment with osimertinib and indicate that it is a resistance mutation to this drug. It seems that only the double mutation p.T790M with the p.C797S in cis is responsible for the resistance
Can the authors clarify this point?
3-Figure 6 B the tracking of the oncogenic EGFRc.2235_2249del15 mutation disappears after treatment with a 1 generation TKI and no other mutations can be detected so the authors conclude on a non-shedder tumour.
In my opinion this qualification is limited by the limit of the number of characterized mutations. In other words, an ultradeep whole exome or genome analysis could refute the absence of circulating tumour DNA.
4-line 354-355 the authors could add the possibility of doing this type of analysis on the cerebrospinal fluid in the case of tumour extension to the central nervous system.
Author Response
The manuscript by De Fernadees MGO et al report a prospective study evaluating the therapeutic value of circulating cell DNA mutation detection in lung cancer. This study, although limited to 13 patients with available tissue and blood samples, shows the value of these analyses. For this purpose, the authors used an NGS approach allowing the characterization of a multitude of targets. Thus, beyond the initial mutations found in the tumour, they can evaluate the disappearance of initial mutations or the appearance of mutations secondary to treatment. Interestingly, the authors indicate that in some cases the characterization of oncogenic mutations may be a more early indication of progression than imaging. The authors also clearly indicate the limits of this characterization and replace these analyses as a new additional element in the patient follow-up system.
Despite the limited number of patients followed, the results are interesting and bring a new light in the management of this disease and therefore deserve to be published with.
Answer: Dear reviewer, thanks so much for your positive comments and interest in our work. Your valuable comments were of utmost importance to further ameriolate the quality of our work.
Minor revisions
1- In the text it would be useful to specify the sensitivity limit retained by the authors for the analyses with the amplicon-based NGS Oncomine lung cfDNA.
Answer: The test sensitivity for detection of both single nucleotide variants (SNVs) and short indels is down to 0.1% LOD. This information was added in materials and methods section.
2-The authors detect in fig5 the appearance of the EGFR p.C797S mutation after treatment with osimertinib and indicate that it is a resistance mutation to this drug. It seems that only the double mutation p.T790M with the p.C797S in cis is responsible for the resistance
Can the authors clarify this point?
Answer: We thank the reviewer for this important observation. We confirm that both mutations were in cis.
3-Figure 6 B the tracking of the oncogenic EGFRc.2235_2249del15 mutation disappears after treatment with a 1 generation TKI and no other mutations can be detected so the authors conclude on a non-shedder tumour.
In my opinion this qualification is limited by the limit of the number of characterized mutations. In other words, an ultradeep whole exome or genome analysis could refute the absence of circulating tumour DNA.
Answer: Thanks for the comment. We absolutely agree with that. With this short-targeted panel, we cannot prove the absence of ctDNA. We changed that designation to “negative”.
4-line 354-355 the authors could add the possibility of doing this type of analysis on the cerebrospinal fluid in the case of tumour extension to the central nervous system.
Answer: Thanks for the suggestion. It was added to the text.
Reviewer 2 Report
The manuscript entitled:" Liquid biopsy for disease monitoring in Non-Small Cell Lung Cancer: the link between biology and the clinic" focused on the evaluation of monitoring by using liquid biopsy in NSCLC patients is well written and requires minor revisions to be accepted for the publication:
- In the material and method section, the authors should clarigy the pre analytical steps for the managment of liquid biopsy samples. In addition, i would also suggest to report the technical parameters inspected for the analysis of blood specimens with NGS platform.
- In the text, please, could the authors report the entire name for the mentioned genes?
- In my opinion, results and discussion section is well documented by the authors. As regards, i would suggest to implement discussion section with the recent paper of Malapelle et al (PMID34115696)
Author Response
The manuscript entitled:" Liquid biopsy for disease monitoring in Non-Small Cell Lung Cancer: the link between biology and the clinic" focused on the evaluation of monitoring by using liquid biopsy in NSCLC patients is well written and requires minor revisions to be accepted for the publication:
Answer: Dear reviewer, thanks so much for your positive comments and interest in our work. Your valuable comments were of utmost importance to further ameriolate the quality of our work.
In the material and method section, the authors should clarify the pre analytical steps for the management of liquid biopsy samples. In addition, I would also suggest to report the technical parameters inspected for the analysis of blood specimens with NGS platform.
Answer: Thanks for your suggestion. Technical details were included in the methods section.
In the text, please, could the authors report the entire name for the mentioned genes?
Answer: Thanks for your suggestion. Definition of the gene names was corrected.
In my opinion, results and discussion section is well documented by the authors. As regards, i would suggest to implement discussion section with the recent paper of Malapelle et al (PMID34115696)
Answer: Dear reviewer, the recently published work of Malapelle et al is extremely interesting and it was considered in the discussion section. Thanks for your tip.